# Improved Analytical Formula for the SAR Doppler Centroid Estimation Standard Deviation for a Dynamic Sea Surface

**Siqi Qiao, Baochang Liu \* and Yijun He** ⓘ

School of Marine Sciences, Nanjing University of Information Science and Technology, Nanjing 210044, China
\* Correspondence: bcliu@nuist.edu.cn

**Abstract:** The existing formulas for the synthetic aperture radar (SAR) Doppler centroid estimation standard deviation (STD) suffer from various limitations, especially for a dynamic sea surface. In this study, we derive an improved version of these formulas through three steps. First, by considering the ocean wavenumber spectrum information, a new strategy for determining the number of independent samples of the sea wave velocity field is adopted in the new formula. This is contrary to the method used in the existing formulas, where the number of SAR geometric resolution cells is taken as the number of samples assuming that adjacent SAR resolution cells are statistically uncorrelated. Second, the pulse repetition frequency and Doppler bandwidth are decoupled in the new formula, unlike in the existing formulas where they are unchangeably related to each other. Third, the effects of thermal noise and Doppler aliasing are jointly quantified in a mathematically exact manner instead of being treated separately, as in the existing formulas. Comprehensive SAR raw data simulations for the ocean surface show that the new formula has a better performance in predicting the Doppler centroid estimation STD than the existing formulas.

**Keywords:** synthetic aperture radar; SAR Doppler centroid estimation standard deviation; analytical model; ocean current retrieval





## 1. Introduction

Ocean currents influence the movement of water on Earth by transporting and mixing salts, gases, heat, and nutrients throughout oceans. Accurate repeated knowledge about the presence, speeds, and directions of ocean currents is crucial to many commercial, societal, and research applications (e.g., oil and gas exploration, route planning for ships, maritime search and rescue, water pollution mapping, and containment [1,2]).

Synthetic aperture radar (SAR) has high resolution and wide coverage and is independent of the operational time of day and weather conditions, thereby proving to be an effective remote sensing tool for measuring ocean surface currents. Generally, two SAR-based methods can be used to measure ocean surface currents. The first method is SAR along-track interferometry (ATI), which measures ocean currents by forming an interferogram based on two complex SAR images of the same scene, but with a short time lag [3–10]. The second method is the Doppler centroid anomaly (DCA) technique. This technique exploits the fact that the difference between a Doppler centroid estimated from SAR data and a theoretical Doppler frequency predicted by the relative motion between a satellite and the rotating Earth reflects the geophysical information about the ocean surface motion. Unlike ATI, which uses two separated antennas (or two split antenna apertures) along the flight track, the DCA technique exploits data from a single antenna, making it applicable to various available SAR datasets without additional preparations [9]. In this paper, we limit our discussion to the relevant issues of the DCA technique when retrieving ocean surface currents.

The use of the DCA technique to measure ocean surface currents was first proposed in the late 1970s [11]. The first demonstration of this technique using European Remote

Sensing Satellite-1 SAR data was reported in 2001 [12]. Since then, the DCA technique has been extensively investigated in several aspects. Since 2005, Chapron et al. [13–15] have applied this technique to Envisat Advanced Synthetic Aperture Radar data and have presented promising results. This technique was also applied to another spaceborne SAR, TerraSAR-X, with the results demonstrating that DCA-derived currents are comparable to short-baseline ATI-derived currents regarding quality [9]. Aside from result demonstration and assessment, algorithm refinement for the DCA technique was studied. For example, Hansen et al. [16] presented the necessary steps for correcting for two measurement bias sources in the geophysical Doppler shift. One bias source was associated with the variation of the normalized radar cross section (NRCS) along azimuth, and the other was linked to an electronic radar beam mispointing and imperfectly known satellite orbit and attitude parameters. Additionally, the expectation value of the radar-detected Doppler velocity from the ocean surface (i.e., the first moment of a random variable) has been theoretically modeled by several researchers [13,17–21]. Johannessen et al. [18] interpreted the mean Doppler velocity as a sum of the radial velocity of ocean surface current, the velocity contribution resulting from tilting and hydrodynamic modulation of scattering facets, and the mean line-of-sight velocity of scattering facets, which is represented as a sum of the phase speed of Bragg waves, advection speed of the specular mirror points, and speed of wave breakers. Apart from theoretical work, an empirical model called the C-band Doppler (CDOP) shift model was developed to characterize the relationship between the wind-induced Doppler shift and the wind speed and direction relative to the radar look direction [22].

In addition to modeling the first-order moment (i.e., mean) of the DCA-derived Doppler velocity, the theoretical modeling of its second-order moment (i.e., statistical fluctuation) is imperative in current-measuring accuracy assessment and error budget computation when designing the instrumental parameters for a new SAR system. Unfortunately, contrary to the comprehensive discussion of its first-order moment modeling, publications on the analytical modeling of the SAR Doppler centroid estimation standard deviation (STD) are relatively few. To our knowledge, only two studies [23,24] have addressed this issue until now. Bamler [23] pioneered the derivation of an analytical expression for the SAR Doppler centroid estimation STD associated with a widely used algorithm, called the average cross-correlation coefficient (ACCC) algorithm [25]. However, Bamler's derivation [23] was performed under the implication of a stationary observation scene, without considering the possible motions of the scene or the quantitative effect of the signal-to-noise ratio (SNR) on the Doppler centroid estimation STD. Liu et al. [24] advanced Bamler's work by considering the properties of a dynamic ocean scene and the effect of the SNR. However, their derived formula for the SAR Doppler centroid estimation STD was still subject to several limitations. First, the effect of ocean surface wave motions was underappreciated in this STD formula because it did not consider the correlation in the large-scale wave motion between two adjacent SAR range resolution cells. Second, in the formula, the pulse repetition frequency (PRF) and Doppler bandwidth were coupled at a fixed oversampling ratio, thereby limiting the use of this formula in practical applications. Third, in deriving this formula, the overall effect of thermal noise and Doppler aliasing on the Doppler centroid estimation STD was quantified as a product of their individual effects, which is rather heuristic instead of being mathematically exact. This limited the ability of the formula to reflect the actual situation. In summary, efforts must be made to improve the generality of modeling the Doppler centroid estimation STD.

This study has two main objectives. The first objective is to derive an improved formula for the SAR Doppler centroid estimation STD of the ACCC algorithm. Thus, efforts are made in three aspects. First, when accounting for the effect of sea wave motions, we adopt a new strategy for determining the number of independent samples of the sea wave velocity field contributing to a Doppler centroid estimate by considering the range correlation length, which is dictated by the sea wavenumber spectrum. This contrasts with the method used in Liu's formula [24], where the number of SAR geometric resolution

cells was simply taken as the number of samples, assuming that adjacent SAR resolution cells were statistically uncorrelated in terms of large-scale ocean wave motion, which is not the case in the actual situation. Second, unlike the formulas derived in previous studies [23,24], where the ratio of the PRF to the Doppler bandwidth was assumed as fixed, these two quantities are decoupled in the newly derived formula such that the azimuthal oversampling ratio can be taken as any value. Third, in deriving the new formula, the effects of thermal noise and Doppler aliasing are jointly quantified in a mathematically exact manner rather than heuristically, as in Liu's formula [24]. The second objective is to perform a validation of the newly derived formula for the Doppler centroid estimation STD against various variables, including radar system and sea state parameters, with the help of SAR raw data simulation for the ocean surface. In addition, we will comprehensively compare the newly derived formula with Bamler's [23] and Liu's [24] formulas to show that the Doppler centroid estimation STDs predicted by the newly derived formula are in a better agreement with the measured values from Monte Carlo simulations than those predicted using Bamler's [23] and Liu's [24] formulas.

The rest of this paper is organized as follows. Section 2 gives a detailed derivation of the improved formula version for the SAR Doppler centroid estimation STD. Section 3 describes a method of Monte Carlo simulations for justifying the validity of the newly derived formula. In Section 4, the new formula is compared with the other existing formulas mentioned earlier. Final conclusions and remarks are provided in Section 5.

## 2. Derivation of an Improved Analytical Formula for the SAR Doppler Centroid Estimation STD for a Dynamic Sea Surface

The SAR Doppler centroid, defined as the Doppler shift of the backscattered radar signal at the antenna beam center, is related to the frequency of the return signal at the radar beam center. This parameter is not only greatly important for SAR data processing (i.e., by affecting the noise and aliasing levels in the processed SAR image [26]) but also useful in measuring geophysical parameters, such as ocean surface currents (Section 1). Due to the azimuth aliasing, the absolute Doppler centroid can be divided into two components: (1) the fractional PRF part that lies in the Doppler interval, ranging from $-0.5 \cdot \text{PRF}$ to $+0.5 \cdot \text{PRF}$, and (2) the integer PRF part. An estimate, $\widehat{f}_{\text{Dc}}$, of the fractional PRF part of the Doppler centroid can be made from received SAR data using the widely used ACCC estimation algorithm as follows [25,26]:

$$\widehat{f}_{\text{Dc}} = \frac{\arg\left\{\sum_k s^*[k] \cdot s[k+1]\right\}}{2\pi} \cdot F_{\text{prf}}, \tag{1}$$

where $F_{\text{prf}}$ denotes the PRF; $s[1], s[2], \cdots, s[k], \cdots$ represent the demodulated azimuth radar signals sampled at a time interval of $1/F_{\text{prf}}$; $(\cdot)^*$ denotes the complex conjugate; and the operator $\arg\{\cdot\}$ means taking the angle of a complex number. Note that the Doppler centroid estimator shown in Equation (1) is equivalent to a correlation-based estimator that works by correlating the observed Doppler power spectrum with a sine function [23].

An analytical formula for the STD of $\widehat{f}_{\text{Dc}}$ estimated by Equation (1) was first derived in [23] for a stationary observation scene. Another version of the formula was developed in [24] to deal with the wave motions of a sea surface. However, the formula in [24] suffered from some drawbacks, including the underappreciation of the effect of ocean wave motions. Here, we derive an improved formula for the Doppler centroid estimation STD obtained by Equation (1) to solve the PRF and Doppler bandwidth decoupling, joint quantification of the effects of Doppler aliasing and thermal noise, and a full expression of the effect of ocean wave motions (Section 1).

For easy derivation, we decompose the overall Doppler centroid estimation variance $\sigma^2_{f_{\mathrm{Dc}}}$ into two parts according to different random fluctuation origins in $\widehat{f}_{\mathrm{Dc}}$:

$$\sigma^2_{f_{\mathrm{Dc}}} = \left(\sigma^{\mathrm{SAR}}_{f_{\mathrm{Dc}}}\right)^2 + \left(\sigma^{\mathrm{sea}}_{f_{\mathrm{Dc}}}\right)^2. \tag{2}$$

where $\left(\sigma^{\mathrm{SAR}}_{f_{\mathrm{Dc}}}\right)^2$ denotes the component of the Doppler centroid estimation variance inherent in the SAR that results from a combined consequence of the multiplicative speckle noise linked to the backscattered radar signals, additive radar receiver thermal noise, along-track antenna beam pattern, and radar motion. $\left(\sigma^{\mathrm{sea}}_{f_{\mathrm{Dc}}}\right)^2$ represents the variance component resulting from the random motions of sea surface waves. Note that the $\sigma^2_{f_{\mathrm{Dc}}}$ partition in Equation (2) assumes that the random process of the fluctuation in $\widehat{f}_{\mathrm{Dc}}$ due to the speckle and thermal noises and the process associated with the ocean surface wave motions are statistically uncorrelated. Additionally, $\left(\sigma^{\mathrm{SAR}}_{f_{\mathrm{Dc}}}\right)^2$ corresponds to a scenario where the radar is in motion but the sea surface is assumed to be stationary. However, $\left(\sigma^{\mathrm{sea}}_{f_{\mathrm{Dc}}}\right)^2$ is related to another scenario where the sea surface is in a random motion state but the radar is located in a stationary fixed position. The following two subsections will give detailed derivations of the analytical formulas for $\left(\sigma^{\mathrm{SAR}}_{f_{\mathrm{Dc}}}\right)^2$ and $\left(\sigma^{\mathrm{sea}}_{f_{\mathrm{Dc}}}\right)^2$.

### 2.1. Derivation of the Formula for $\left(\sigma^{SAR}_{f_{Dc}}\right)^2$

Presently, we will assume a moving radar but a stationary sea surface. To begin with, we repeat a formula derived in [23] for the ACCC-algorithm-associated Doppler centroid estimation variance component pertaining to the speckle and thermal noises (Equation (36), [23]):

$$\left(\sigma^{\mathrm{SAR}}_{f_{\mathrm{Dc}}}\right)^2 = \frac{F^2_{\mathrm{prf}} \cdot \gamma^{\mathrm{rg}}_{\mathrm{osr}}}{N_p \cdot N_r} \cdot \frac{1}{2\pi^2} \cdot \left(\frac{1}{m^2} + \frac{1}{4}\right), \tag{3}$$

where $N_p$ is the number of pulses contributing to the Doppler centroid estimate; $N_r$ is the number of range samples used to estimate the Doppler centroid using the ACCC algorithm; $\gamma^{\mathrm{rg}}_{\mathrm{osr}}$ is the range oversampling ratio; and $m$, called the spectrum sharpness factor, characterizes the degree of sharpness of the expected Doppler power spectrum of the received SAR data, which is defined according to [23] as follows:

$$m = \frac{P_{\mathrm{max}} - P_{\mathrm{min}}}{P_{\mathrm{max}} + P_{\mathrm{min}}}, \tag{4}$$

where $P_{\mathrm{max}}$ and $P_{\mathrm{min}}$ represent the maximum and the minimum of the expected Doppler power spectrum, respectively (Figure 1).

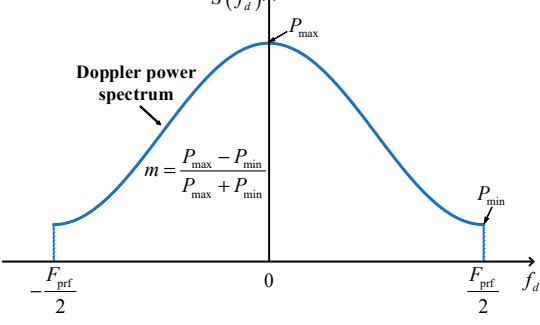

**Figure 1.** Doppler power spectrum of the SAR and the spectrum sharpness factor $m$.

Note that the expression for $\left(\sigma_{f_{\mathrm{Dc}}}^{\mathrm{SAR}}\right)^2$ shown in Equation (3) was derived assuming a stationary observation scene and with the expected Doppler power spectrum of the received SAR data $S_{\mathrm{SAR}}(f_d)$ taking on the shape of a cosine wave on a pedestal [23]:

$$S_{\mathrm{SAR}}(f_d) = 1 + m \cdot \cos\left(2\pi f_d / F_{\mathrm{prf}}\right), \tag{5}$$

where $f_d$ denotes the Doppler frequency variable, and $m$ is the spectrum sharpness factor, as in Equation (4). Note that this cosine-shaped $S_{\mathrm{SAR}}(f_d)$ is well approximated by the first-order Fourier series of a sinc-quartic function [23]. In addition, the Doppler centroid estimator using the ACCC algorithm shown in Equation (1) works by correlating the observed Doppler power spectrum with a sine function [23], meaning that the choice of the cosine-shaped Doppler spectral model is applicable. Equation (3) was derived assuming that the PRF and the SAR Doppler bandwidth relate to each other at a fixed azimuth oversampling ratio slightly larger than one, thereby limiting the application scope of this formula. Subsequently, we attempt to solve this problem by extending the formula in Equation (3) to the case where the PRF and Doppler bandwidth can take any value.

Let us consider a scenario where the PRF, $F_{\mathrm{prf}}$, is much larger than the SAR Doppler bandwidth $B_D$. Here, two contiguous azimuth samples of the received SAR raw data are not fully statistically independent of each other, meaning that redundancy occurs among the azimuth signal samples. To eliminate this redundancy, we subsample the azimuth SAR data by a factor $F_{\mathrm{prf}}/B_D$ such that the equivalent PRF of the subsampled SAR data becomes $F'_{\mathrm{prf}} = B_D$, which is smaller than the original PRF, $F_{\mathrm{prf}}$. After subsampling, the effective number of independent azimuth samples contributing to the Doppler centroid estimate is reduced from its original number $N_p$ to a smaller number $N_p^{\mathrm{eff}}$, expressed as follows:

$$N_p^{\mathrm{eff}} = N_p \cdot \frac{B_D}{F_{\mathrm{prf}}} = T_{\mathrm{az}} \cdot B_D, \tag{6}$$

where $T_{\mathrm{az}} = N_p / F_{\mathrm{prf}}$ is the azimuth observation time contributing to the Doppler centroid estimate. Replacing $F_{\mathrm{prf}}$ and $N_p$ in Equation (3) with $F'_{\mathrm{prf}} = B_D$ and $N_p^{\mathrm{eff}}$, respectively, we have the following:

$$\left(\sigma_{f_{\mathrm{Dc}}}^{\mathrm{SAR}}\right)^2 = \frac{B_D \cdot \gamma_{\mathrm{osr}}^{\mathrm{rg}}}{T_{\mathrm{az}} \cdot N_r} \cdot \frac{1}{2\pi^2} \cdot \left(\frac{1}{m^2} + \frac{1}{4}\right). \tag{7}$$

From Equation (7), we propose the following comments on the implications of the $\left(\sigma_{f_{\mathrm{Dc}}}^{\mathrm{SAR}}\right)^2$ expression:

- Expressing $\left(\sigma_{f_{\mathrm{Dc}}}^{\mathrm{SAR}}\right)^2$ as a function of $B_D$ and $T_{\mathrm{az}}$, rather than PRF, has the consequence of decoupling the PRF and the Doppler bandwidth because they can each take any value independently.

- How the four parameters of $B_D$, $T_{\mathrm{az}}$, $m$ and $N_r$ in Equation (7) govern the $\left(\sigma_{f_{\mathrm{Dc}}}^{\mathrm{SAR}}\right)^2$ value is explained as follows:

  1. Given that the Doppler centroid estimator shown in Equation (1) is equivalent to the Doppler centroid estimator that works by correlating the Doppler power spectrum of the received SAR data with a sine function, the total amount of noise incorporated into the correlation results reduces with a decrease in the Doppler bandwidth; thus, the smaller the value of $B_D$ (or what is equivalent, the narrower the radar beam in the azimuth direction), the lower the value of $\left(\sigma_{f_{\mathrm{Dc}}}^{\mathrm{SAR}}\right)^2$.

  2. An increase in the amount of the scene observation time, increases the accuracy of the Doppler centroid measurement; hence, the larger the value of $T_{\mathrm{az}}$, the smaller the value of $\left(\sigma_{f_{\mathrm{Dc}}}^{\mathrm{SAR}}\right)^2$.

3. An increase in the sharpness of the shape that the Doppler power spectrum exhibits facilitates the determination of the Doppler centroid in the presence of speckle and thermal noises. Therefore, the larger the spectrum sharpness factor $m$, the smaller the value of $\left(\sigma_{f_{\text{Dc}}}^{\text{SAR}}\right)^2$.

4. It is a straightforward fact that the larger the number of range samples, $N_r$, the smaller the value of $\left(\sigma_{f_{\text{Dc}}}^{\text{SAR}}\right)^2$.

After $\left(\sigma_{f_{\text{Dc}}}^{\text{SAR}}\right)^2$ is expressed as a function of the SAR Doppler bandwidth and the azimuth observation time, the issue of deriving an analytical expression for the spectrum sharpness factor $m$ involved in Equation (7) remains to be addressed. In the relevant formula in [23] (Equation (36), [23]) or Equation (3) in this paper), the spectrum sharpness factor was set to a fixed number ($m = 0.7$) without considering its variations with the Doppler aliasing and receiver thermal noise levels. The effect of thermal noise on $m$ was considered in Liu's formula (Equation (A5), [24]). However, in this formula, the overall effect of the thermal noise and Doppler aliasing on $m$ was quantified as a product of their individual effects, a rather coarse method. Thus, the question of how to obtain the expression for $m$ in a mathematically exact manner arises.

To deal with the abovementioned problem, we first consider an ideal case where no Doppler aliasing occurs. For this case, the expected Doppler power spectrum of the radar echoes can be expressed as follows, assuming a sinc-squared two-way antenna beam pattern:

$$S_{\text{una}}(f_d) = P_0 \cdot \text{sin c}^4\left(\frac{f_d}{B_D}\right), \tag{8}$$

where the subscript "una" indicates the unaliased power spectrum, $\text{sin c}(x) = \sin(\pi x)/(\pi x)$ is the sinc function, and the constant $P_0$ represents the maximum value of $S_{\text{una}}(f_d)$. As most SAR antennas are unweighted along the azimuth direction, the one-way beam pattern is approximately a sinc function so that the two-way beam pattern is a sinc-squared function. Due to the envelope of the SAR Doppler signal having the same shape with the two-way beam pattern, the expexted Doppler power spectrum of the radar echoes, which is the square amplitude of the SAR Doppler signal, can be expressed as a sinc-quartic function. Note that, as a starting point for deriving an analytical expression for the spectrum sharpness factor $m$, $S_{\text{una}}(f_d)$ can be expressed as a sinc-quartic function without requiring a cosine-shaped approximation. The plot of $S_{\text{una}}(f_d)$ versus $f_d$ is drawn as a solid line in Figure 2. However, in practical situations, a certain degree of Doppler aliasing is always present in the observed Doppler power spectrum due to the discrete sampling of the azimuth SAR signal and the azimuth signal not being band limited. Therefore, the Doppler spectral components that lie outside the Doppler interval $\left[-F_{\text{prf}}/2, +F_{\text{prf}}/2\right]$ will be folded into this interval, as illustrated by the dashed lines in Figure 2. By referring to Figure 2 and considering only first-order ambiguities, we can easily express the aliased part of the Doppler power spectrum $S_{\text{ali}}(f_d)$ as follows:

$$S_{\text{ali}}(f_d) = S_{\text{una}}\left(f_d + F_{\text{prf}}\right) + S_{\text{una}}\left(f_d - F_{\text{prf}}\right), \quad -\frac{F_{\text{prf}}}{2} \leq f_d \leq \frac{F_{\text{prf}}}{2}. \tag{9}$$

The thermal noise of the radar receiver is independent of the complex signal of the radar echo, and the complex unaliased and aliased signal components are statistically uncorrelated. Hence, the total observed SAR power spectrum $S_{\text{SAR}}(f_d)$, defined over the Doppler interval $\left[-F_{\text{prf}}/2, +F_{\text{prf}}/2\right]$, can be written as the sum of the unaliased spectrum part, aliased spectrum part, and thermal noise spectrum:

$$S_{\text{SAR}}(f_d) = S_{\text{una}}(f_d) + S_{\text{ali}}(f_d) + P_n, \quad -\frac{F_{\text{prf}}}{2} \leq f_d \leq \frac{F_{\text{prf}}}{2}, \tag{10}$$

where $P_n$ is the expected power spectrum of the thermal noise, for which the thermal noise has been considered to be white with a power spectral density proportional to a fixed value [23]. The $S_{\text{SAR}}(f_d)$ curve is represented by the dash–dotted line in Figure 2.

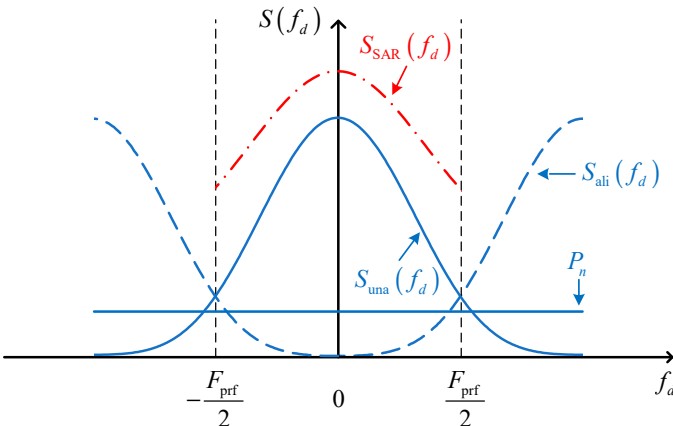

**Figure 2.** Doppler power spectra with and without Doppler aliasing. The solid, dashed, and dash–dotted lines plot the unaliased Doppler power spectrum, aliased part of the Doppler power spectrum, and total observed Doppler power spectrum, respectively. The receiver thermal noise power spectrum is drawn as a horizontal line.

Using Equation (4) and referring to Figure 1, the spectrum sharpness factor $m$ can be expressed as follows:

$$m = \frac{\max\{S_{\text{SAR}}(f_d)\} - \min\{S_{\text{SAR}}(f_d)\}}{\max\{S_{\text{SAR}}(f_d)\} + \min\{S_{\text{SAR}}(f_d)\}} = \frac{S_{\text{SAR}}(0) - S_{\text{SAR}}\left(F_{\text{prf}}/2\right)}{S_{\text{SAR}}(0) + S_{\text{SAR}}\left(F_{\text{prf}}/2\right)}. \tag{11}$$

Substituting Equations (8)–(10) into (11), defining the azimuth oversampling ratio as $\gamma_{\text{osr}}^{\text{az}} = F_{\text{prf}}/B_D$ and the SNR as $\text{SNR} = (P_0/2)/P_n$, and performing some trivial algebraic manipulations, we obtain the analytical formula for $m$ as a function of the azimuth oversampling ratio $\gamma_{\text{osr}}^{\text{az}}$ and the SNR:

$$m(\gamma_{\text{osr}}^{\text{az}}, \text{SNR}) = \frac{1 - 2 \cdot \text{sin c}^4(0.5 \cdot \gamma_{\text{osr}}^{\text{az}}) + 2 \cdot \text{sin c}^4(\gamma_{\text{osr}}^{\text{az}}) - \text{sin c}^4(1.5 \cdot \gamma_{\text{osr}}^{\text{az}})}{1 + 2 \cdot \text{sin c}^4(0.5 \cdot \gamma_{\text{osr}}^{\text{az}}) + 2 \cdot \text{sin c}^4(\gamma_{\text{osr}}^{\text{az}}) + \text{sin c}^4(1.5 \cdot \gamma_{\text{osr}}^{\text{az}}) + \frac{1}{\text{SNR}}}. \tag{12}$$

Substituting the relevant radar system and configuration parameters into $\gamma_{\text{osr}}^{\text{az}} = F_{\text{prf}}/B_D$, we can further express $\gamma_{\text{osr}}^{\text{az}}$ as follows [24]:

$$\gamma_{\text{osr}}^{\text{az}} = \frac{F_{\text{prf}} \cdot D_a}{1.772 \cdot v_s \cdot \alpha_b^t \cdot \alpha_b^r}, \tag{13}$$

where $D_a$ is the along-track antenna length; $v_s$ is the effective radar velocity; and $\alpha_b^t$ and $\alpha_b^r$ denote the beam broadening factors on transmit and receive, respectively. Relating the SNR to the ocean surface environmental parameters, we have the following equation:

$$\text{SNR} = \frac{\sigma^0(U, \phi_{\text{wind}}, \theta_{\text{inc}}, \text{pol})}{\sigma_{\text{NE}}^0}, \tag{14}$$

where $\sigma_{\text{NE}}^0$ is the noise equivalent sigma zero (NESZ), an important radar system parameter characterizing the radiometric sensitivity of a SAR system, $\sigma^0$ is the NRCS, which is a function of the wind speed at a height of 10 m, the wind direction angle $\phi_{\text{wind}}$ relative to the radar look direction, the incidence angle $\theta_{\text{inc}}$ of the radar beam, and the polarization pol.

Using $B_D = 1.772 \cdot v_s \cdot \alpha_b^t \cdot \alpha_b^r / D_a$ and inserting Equation (12) into Equation (7), we can obtain the final form of $\left(\sigma_{f_{\text{Dc}}}^{\text{SAR}}\right)^2$ as follows:

$$\left(\sigma_{f_{\text{Dc}}}^{\text{SAR}}\right)^2 = \frac{0.886 \cdot v_s \cdot \alpha_b^t \cdot \alpha_b^r \cdot \gamma_{\text{osr}}^{\text{rg}}}{\pi^2 \cdot D_a \cdot T_{\text{az}} \cdot N_r} \cdot \left(\frac{1}{m^2(\gamma_{\text{osr}}^{\text{az}}, \text{SNR})} + \frac{1}{4}\right), \tag{15}$$

where the specific expression for $m^2(\gamma_{\text{osr}}^{\text{az}}, \text{SNR})$ can be found in Equations (12)–(14). According to the specific expression for the Doppler bandwidth, we find that the Doppler bandwidth is proportional to the effective radar velocity with all the other parameters constant; on the other hand, the $\left(\sigma_{f_{\text{Dc}}}^{\text{SAR}}\right)^2$ value is related to the Doppler bandwidth. Therefore, the effect of the radar motion on $\left(\sigma_{f_{\text{Dc}}}^{\text{SAR}}\right)^2$ can be explained as follows: the Doppler bandwidth reduces with a reduction in the effective radar velocity. Thus, the total amount of noise incorporated into the correlation results reduces, so that the spectrum sharpness factor increases. A combined consequence of the aforementioned processes induces the decrease in $\left(\sigma_{f_{\text{Dc}}}^{\text{SAR}}\right)^2$.

### 2.2. Derivation of the Formula for $\left(\sigma_{f_{\text{Dc}}}^{sea}\right)^2$

In deriving the formula for $\left(\sigma_{f_{\text{Dc}}}^{\text{SAR}}\right)^2$ in Section 2.1, the equivalent number of independent range samples averaged to obtain the Doppler centroid estimate is taken as $N_r / \gamma_{\text{osr}}^{\text{rg}}$ [Equation (15)], which is the number of geometric range resolution cells. This is a valid treatment because random noise processes, including the speckle and thermal noises, in two different range resolution cells are statistically uncorrelated. In Liu's formula (Equation (A8), [24]), the effect of the sea surface motions was modeled as a broadening of the SAR Doppler power spectrum before being directly combined with the noise effect. That is, the processes of the sea surface motion in two adjacent range resolution cells were independent of each other in their study, just as the speckle and thermal noises are treated. This means that in Liu's formula (Equation (A1), [24]), the number of independent range samples of the sea wave velocity field was taken as $N_r / \gamma_{\text{osr}}^{\text{rg}}$, similar to that for the noise process. However, in practical situations, doing so will underappreciate the effect of the sea wave motions on the Doppler centroid estimation variance because the sea wave orbital velocities on the sea surface always exhibit some patterns, instead of showing a white-noise-like feature. Hence, the number of independent range samples of the sea wave velocity field may be less than $N_r / \gamma_{\text{osr}}^{\text{rg}}$. In what follows, we will give a detailed derivation of the expression for $\left(\sigma_{f_{\text{Dc}}}^{\text{sea}}\right)^2$ shown in Equation (2) using a new strategy, focusing on how to determine the number of independent range samples of the sea wave velocity field.

To proceed with our derivations, we assume a scenario where the radar velocity is set to zero, but the sea surface is in a dynamic state. We then consider a narrow strip on the sea surface aligned parallel to the azimuth direction, with a length corresponding to the azimuth size of the antenna beam footprint and a width the size of a single range resolution cell.

The general assumption that the sea surface can be described as a superposition of a series of free sine waves with independent phases yields the fact that the sea wave velocity field, $\mathbf{V_R}(x, y)$, where $x$ and $y$ denote the azimuth and range coordinates, respectively, can also be expressed as a superposition of a group of free sine functions [17]. This results in the Doppler power spectrum of the moving sea surface confined within the aforementioned narrow strip being represented by the probability density of a Gaussian-distributed sea wave velocity field:

$$S_{\text{sea}}(f_d) = a_0 \cdot \exp\left\{-\frac{1}{2} \cdot \left(\frac{f_d - \bar{f}_d}{B_D^{\text{sea}}}\right)^2\right\}, \tag{16}$$

where $a_0$ is proportionality factor, $\overline{f}_d$ is the mean Doppler shift of the moving sea surface, and $B_D^{\text{sea}}$ denotes the Doppler bandwidth of the moving sea surface.

Provided that the shape of the Doppler power spectrum of the moving sea surface given in Equation (16) is highly similar to that of the Doppler power spectrum of the SAR in Equation (5), we can resort to the same formula [Equation (7)] employed for computing $\left(\sigma_{f_{\text{Dc}}}^{\text{SAR}}\right)^2$ to evaluate the component of the Doppler centroid estimation variance $\left(\sigma_{f_{\text{Dc}}}^{\text{sea}}\right)^2$ linked to the random sea surface motions by substituting the spectrum sharpness factor, Doppler bandwidth of the sea surface, and the number of independent range samples of the sea wave velocity field $\mathbf{V_R}(x,y)$ into Equation (7). We then obtain the following:

$$\left(\sigma_{f_{\text{Dc}}}^{\text{sea}}\right)^2 = \frac{B_D^{\text{sea}}}{T_{\text{az}} \cdot N_r^{\text{sea}}} \cdot \frac{1}{2\pi^2} \cdot \left(\frac{1}{m_{\text{sea}}^2} + \frac{1}{4}\right), \tag{17}$$

where $m_{\text{sea}}$ and $N_r^{\text{sea}}$ are the spectrum sharpness factor of the Doppler spectrum of the sea surface and the number of independent range samples of the sea wave velocity field, respectively, and $T_{\text{az}}$ is the radar observation time, which has the same value as that in Equation (7). Thus, our task here is to obtain specific analytical expressions for $m_{\text{sea}}$, $B_D^{\text{sea}}$, and $N_r^{\text{sea}}$.

The PRF of a radar is typically far greater than the Doppler bandwidth of the sea surface; therefore, the $S_{\text{sea}}(f_d)$ value at $f_d = \pm F_{\text{prf}}/2$ is approximately zero (i.e., $S_{\text{sea}}\left(\pm F_{\text{prf}}/2\right) \approx 0$). According to the definition of the spectrum sharpness factor given in Equation (4), $m_{\text{sea}}$ is evaluated as follows:

$$\begin{aligned} m_{\text{sea}} &= \frac{\max\{S_{\text{sea}}(f_d)\} - \min\{S_{\text{sea}}(f_d)\}}{\max\{S_{\text{sea}}(f_d)\} + \min\{S_{\text{sea}}(f_d)\}} \\ &= \frac{S_{\text{sea}}(0) - S_{\text{sea}}\left(\pm F_{\text{prf}}/2\right)}{S_{\text{sea}}(0) + S_{\text{sea}}\left(\pm F_{\text{prf}}/2\right)} \\ &\approx 1 \end{aligned} \tag{18}$$

By definition, the Doppler bandwidth $B_D^{\text{sea}}$ of the sea surface is related to the root-mean-square (RMS) radial velocity $\sigma_{v_r}^{\text{sea}}$ of large-scale gravity sea waves (contrary to small-scale Bragg scattering waves) as follows:

$$B_D^{\text{sea}} = \frac{2 \cdot \sigma_{v_r}^{\text{sea}}}{\lambda}, \tag{19}$$

where $\lambda$ is the radar wavelength. The RMS radial velocity $\sigma_{v_r}^{\text{sea}}$, according to [27,28], is evaluated as follows:

$$\sigma_{v_r}^{\text{sea}} = \sqrt{\int \left\{ \left[\cos^2(\theta_{\text{inc}}) + \sin^2(\theta_{\text{inc}}) \cdot \cos^2(\varphi(\mathbf{k}))\right] \cdot \omega^2(\mathbf{k}) \cdot F_{\text{wave}}(\mathbf{k}) \right\} d\mathbf{k}}, \tag{20}$$

where $\theta_{\text{inc}}$ is the incidence angle of the radar; $\mathbf{k}$ is the sea wavenumber vector; $F_{\text{wave}}(\mathbf{k})$ is the sea wave height spectrum; $\omega$ is the angular frequency of the sea wave, whose wavenumber vector is $\mathbf{k}$; and $\varphi$ is the relative angle between the sea wave propagation and radar look directions. Note that both $\omega$ and $\varphi$ are functions of the sea wavenumber vector $\mathbf{k}$. Furthermore, the integration in Equation (20) is performed for wavenumbers up to one-sixth of the Bragg wavenumber.

If we further assume a fully developed wind sea, the wavenumber spectrum $F_{\text{wave}}(\mathbf{k})$ in Equation (20) will take on the form of the well-known Pierson–Moskowitz (PM) spectrum [29–31], $F_{\text{PM}}(k,\phi)$:

$$F_{\text{PM}}(k,\phi) = \frac{0.016}{3\pi} k^{-4} \cdot \exp\left[-\frac{5}{4}\left(\frac{k_{\text{peak}}}{k}\right)^2\right] \cdot \cos^4\left(\phi - \phi_{\text{peak}}\right), \tag{21}$$

where $k$ and $\phi$ are the magnitude and the propagation direction angle of the wavenumber vector $\mathbf{k}$, respectively, and $k_{\text{peak}}$ and $\phi_{\text{peak}}$ are the peak wavenumber of the PM spectrum

and the propagation direction angle of the peak sea wave component, respectively. $k_{\text{peak}}$ is expressed as follows [29,31]:

$$k_{\text{peak}} = 0.7 \frac{g}{U^2}, \tag{22}$$

where $g$ is the gravity acceleration, and $U$ is the wind speed. Typically, the newly derived formula for the SAR Doppler centroid estimation STD must be computationally efficient and valid. Regarding efficiency, by inserting Equations (21) and (22) into Equation (20), we can approximate the expression for $\sigma_{v_r}^{\text{sea}}$ to a rather simple form [28], which is simply a function of the wind speed $U$:

$$\sigma_{v_r}^{\text{sea}}(U) \approx \frac{U}{6\sqrt{2\pi}}. \tag{23}$$

To demonstrate the validity of the approximation in Equation (23), Figure 3 plots the curves of the RMS radial velocity variation against the wind speed and the relative angle between the sea wave propagation and radar look directions, which are computed using Equations (20) and (23), respectively. Figure 3a shows that the curve of the RMS radial velocity computed using Equation (23) is in good agreement with that obtained from Equation (20). Additionally, their variation trend against the wind speed is shown to be significant, indicating that the approximate formula effectively characterizes the changes in the RMS radial velocity against the wind speed. In contrast, as shown in Figure 3b, the extent of the change in the RMS radial velocity computed using Equation (23) versus the relative angle is relatively small compared with that of the approximate formula given in Equation (23).

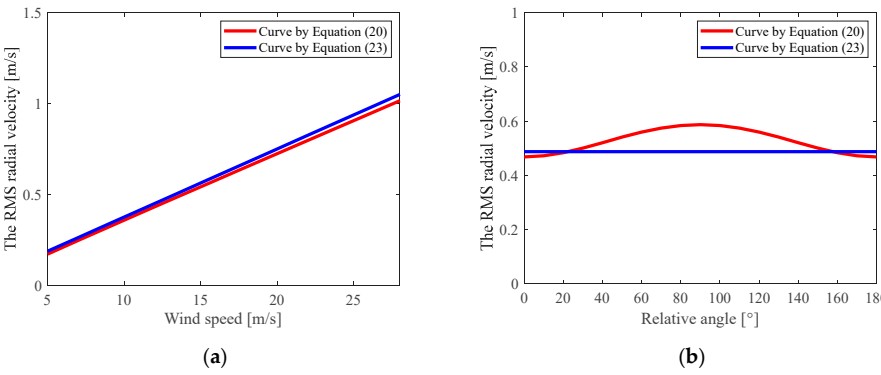

**Figure 3.** Curves of the RMS radial velocity versus (**a**) the wind speed; (**b**) the relative angle between the sea wave propagation and radar look directions.

Substituting Equation (23) into Equation (19) establishes the following relationship between the Doppler bandwidth $B_D^{\text{sea}}$ of the sea surface and the geophysical parameter $U$:

$$B_D^{\text{sea}}(U, \lambda) \approx \frac{U}{3\sqrt{2\pi} \cdot \lambda}. \tag{24}$$

Having derived the specific expressions for $m_{\text{sea}}$ and $B_D^{\text{sea}}$, we now have only one parameter involved in Equation (17) (i.e., $N_r^{\text{sea}}$) whose specific expression remains to be derived. As discussed in Section 2.2, the effective number of independent range samples of the sea wave velocity field $\mathbf{V_R}(x, y)$ should not be simply taken as the number of geometric range resolution cells. Rather, the degree of spatial correlation between two different locations on the sea wave velocity field should be considered when deriving the specific expression for $N_r^{\text{sea}}$. Following this consideration, $N_r^{\text{sea}}$ can be expressed as follows:

$$N_r^{\text{sea}} = \frac{L_y}{l_{\text{cor}}}, \tag{25}$$

where $L_y$ is the ground-range length of a sea surface region from which one single SAR Doppler centroid estimate is obtained, $l_{cor}$ is the correlation length of the sea wave velocity field defined such that if the spacing between two locations is larger than $l_{cor}$, then the velocities at both locations are considered statistically uncorrelated. $l_{cor}$ is evaluated following the signal theory stating that the power spectral density of a random signal is the Fourier transform of the autocorrelation function of this signal:

$$l_{cor} = \frac{2\pi}{\Delta k}, \tag{26}$$

where $\Delta k$ is the spectral width of the wavenumber power spectrum $F_{VF}(k, \phi)$ of the sea wave velocity field $\mathbf{V_R}(x, y)$. The larger the value of $\Delta k$ the more the sea wave velocity field distribution tends to be white noise, and thus the smaller the correlation length $l_{cor}$ of the sea wave velocity field, as shown in Equation (26).

Next, we derive the analytical expression for $\Delta k$. To do this, we still consider the case of a fully developed wind sea. Using the relationship between the wave heights of large-scale waves and their corresponding orbital velocities [17,28,32] and considering Equation (20) again, we can relate $F_{VF}(k, \phi)$ to the wave height spectrum $F_{PM}(k, \phi)$ as follows:

$$F_{VF}(k, \phi) = \left[ \cos^2(\theta_{inc}) + \sin^2(\theta_{inc}) \cdot \cos^2(\varphi) \right] \cdot \omega^2 \cdot F_{PM}(k, \phi). \tag{27}$$

Subsequently, the wavenumber spectral width $\Delta k$ of $F_{VF}(k, \phi)$ can be equivalently determined as the width of a two-dimensional (2D) rectangle function with a height equal to the maximum value of $F_{VF}(k, \phi)$ and a volume equal to that enclosed between the surface of $F_{VF}(k, \phi)$ and the 2D wavenumber-axis plane, as shown in Figure 4. If treated mathematically, $\Delta k$ is determined as follows:

$$\Delta k = \sqrt{\frac{\iint F_{VF}(k, \phi) k \, dk \, d\phi}{\max\{F_{VF}(k, \phi)\}}}. \tag{28}$$

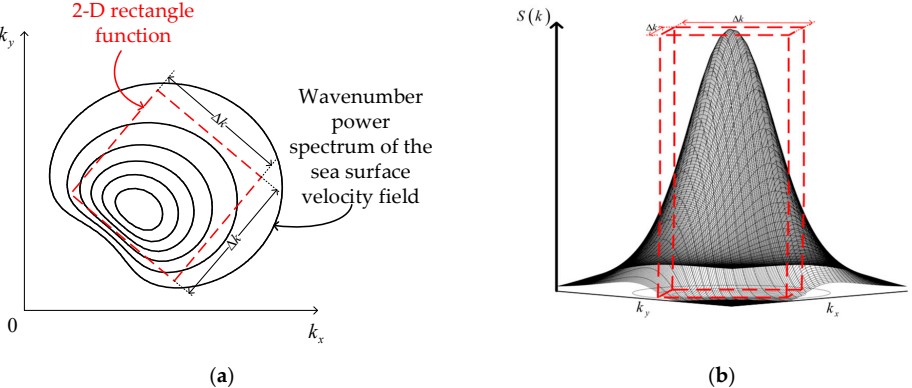

**Figure 4.** Illustration of the wavenumber power spectrum of the sea wave velocity field and its equivalent spectral width. (**a**) Top view; (**b**) Side view.

A numerical computation shows that $\Delta k$ in Equation (28) can be approximated by the following equation:

$$\begin{aligned} \Delta k &\approx 1.87 \cdot k_{peak} \\ &\approx \frac{1.31 \cdot g}{U^2} \end{aligned}. \tag{29}$$

To demonstrate the validity of the approximation in Equation (29), Figure 5 plots the curve of the $\Delta k$ variation against the wind speed $U$ computed using Equation (28) and that obtained by the approximate formula given in Equation (29); as can be seen, they agree excellently with each other. Equation (29) shows that the wavenumber spectral width $\Delta k$ of

$F_{\text{VF}}(k, \phi)$ is directly proportional to the peak wavenumber $k_{\text{peak}}$ and inversely proportional to the squared wind speed $U^2$. Substituting Equation (29) into Equation (26) gives the following:

$$l_{\text{cor}} \approx \frac{2\pi}{1.31 \cdot g} U^2. \tag{30}$$

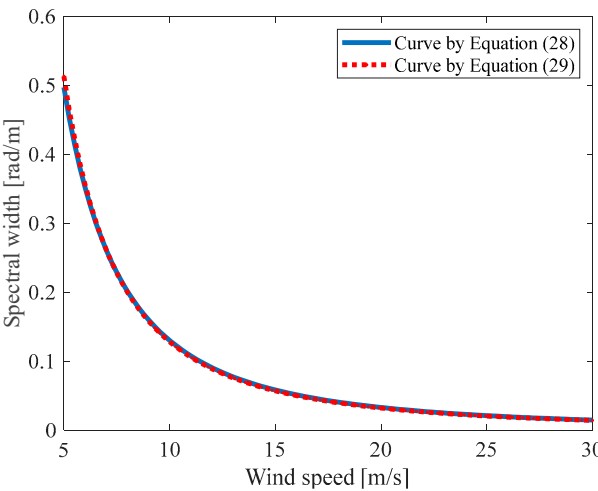

**Figure 5.** Curves of $\Delta k$ versus $U$ computed by Equations (28) and (29).

It is found from Equation (30) that the correlation length $l_{\text{cor}}$ of the radial sea wave velocity field $\mathbf{V_R}(x, y)$ is proportional to the square of the wind speed $U^2$.

Let us now return to Equation (25), where $L_y$ can be expressed as follows:

$$L_y = N_r \cdot \frac{c}{2F_s \cdot \sin(\theta_{\text{inc}})}, \tag{31}$$

where $N_r$ is the number of range samples of the SAR data. Inserting Equations (30) and (31) into Equation (25) yields the following equation:

$$N_r^{\text{sea}} \approx N_r \cdot \frac{1.31 \cdot g \cdot c}{4\pi \cdot F_s \cdot \sin(\theta_{\text{inc}})} \cdot \frac{1}{U^2}. \tag{32}$$

From Equation (32), the effective number of independent range samples of the sea wave velocity field is found to be inversely proportional to the squared wind speed $U^2$. Additionally, Equation (32) shows that the effective number of independent range samples of the sea surface velocity field can be expressed as $N_r$ multiplied by a factor, $1.31 \cdot g \cdot c / [4\pi \cdot F_s \cdot \sin(\theta_{\text{inc}}) \cdot U^2]$, which is usually considerably smaller than one for typical spaceborne SAR system parameters and sea wind speeds ranging from 5 to 28 m/s. This indicates the significance of the effect of the correlation between two locations of the sea surface velocity field on the Doppler centroid estimation variance.

Substituting Equations (18), (24), and (32) into Equation (17) yields the following analytical expression for $\left(\sigma_{f_{\text{Dc}}}^{\text{sea}}\right)^2$:

$$\left(\sigma_{f_{\text{Dc}}}^{\text{sea}}\right)^2 \approx \frac{0.636}{\sqrt{2}\pi^2 g} \cdot \frac{F_s \cdot \sin(\theta_{\text{inc}})}{T_{\text{az}} \cdot \lambda \cdot c \cdot N_r} \cdot U^3. \tag{33}$$

Interestingly, the component of the Doppler centroid estimation variance originating from the ocean surface wave motions is proportional to the third power of the wind speed $U^3$.

In addition, the overall STD $\sigma_{f_{\text{Dc}}}$, of the SAR Doppler centroid estimate is expressed as follows:

$$\sigma_{f_{\text{Dc}}} = \sqrt{\left(\sigma_{f_{\text{Dc}}}^{\text{SAR}}\right)^2 + \left(\sigma_{f_{\text{Dc}}}^{\text{sea}}\right)^2}. \tag{34}$$

Here, the specific expression for $\left(\sigma_{f_{\text{Dc}}}^{\text{SAR}}\right)^2$ is presented in Equations (12)–(15), whereas that for $\left(\sigma_{f_{\text{Dc}}}^{\text{sea}}\right)^2$ is given in Equation (33).

## 3. Method of Monte Carlo Simulations

To justify the effectiveness of the newly derived formula [Equation (34)] in characterizing the SAR Doppler centroid estimation STD for a moving sea surface, we consider using the method of Monte Carlo simulations. In this section, we describe the main procedures of this method. Section 4 will provide a detailed discussion on Monte Carlo simulation results obtained from varying radar system and sea state parameters.

### 3.1. Procedures of Monte Carlo Simulations

Figure 6 presents a flowchart of Monte Carlo simulations used in this study. The flowchart generally comprises the steps of SAR raw data simulation, Doppler centroid estimation, and statistical analysis. The details of each step in this flowchart are provided below.

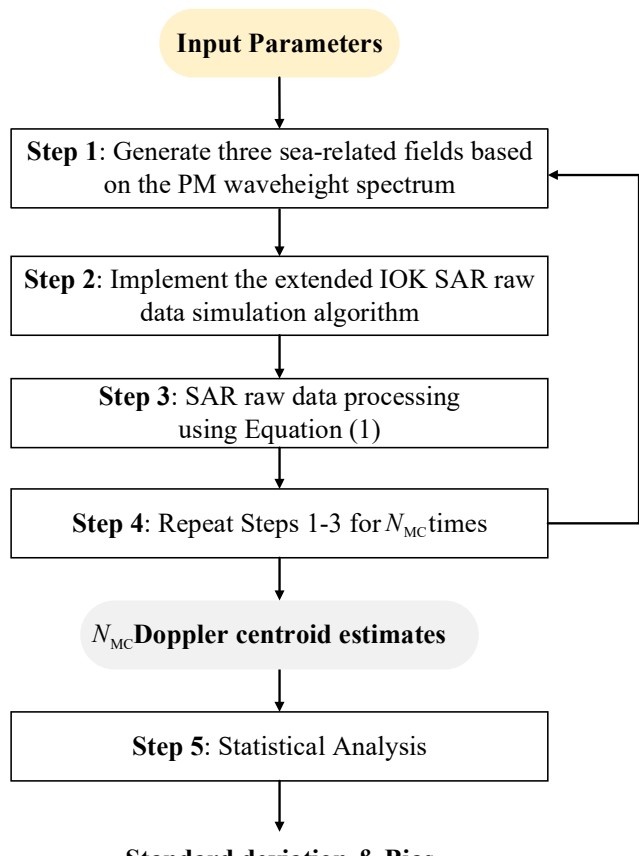

**Figure 6.** Flowchart of Monte Carlo simulations.

The flowchart begins with a prescription of the input parameters, including the radar system and sea environmental parameters, which are then employed to obtain a specific PM wave height spectrum using Equation (21), as described in Step 1 in Figure 6. From the resulting PM spectrum, we can generate a 2D wave height field, a 2D NRCS field, and a 2D radial sea wave velocity field. The following explanations are specifically given:

- A 2D random discrete wave height field with a given grid cell spacing is generated as a superimposition of a series of 2D sine harmonic waves (As discussed in [31], at wavelengths more than 10 m in our simulation, the interactions between small- and large-scale waves can be considered negligible. Consequently, the sea surface can be defined by a Gaussian probability density function where the phases of each

component are independent and equally distributed between 0 and $2\pi$.), with each harmonic wave component characterized by its amplitude, phase, and propagation direction [31]. The amplitude of each harmonic wave component is generated by a Rayleigh distribution random number generator with a variance set to the energy contained in each spectral bin of the resulting PM wave height spectrum. The phase of each harmonic wave component is generated by a random number generator equally distributed between 0 and $2\pi$. The phases of the individual harmonic wave components are made to be statistically uncorrelated to obtain a Gaussian-distributed sea surface.

- Once a 2D wave height field is realized, we can compute the local incidence angle at each grid cell on the generated discrete 2D sea surface. Subsequently, these local incidence angles are used as input parameters in a theoretical formula for the NRCS in the Bragg regime [33] such that a 2D discrete NRCS field can be realized. Hydrodynamic modulation is also considered in generating the NRCS field. Afterward, the 2D random discrete complex reflectivity field can be generated using a Gaussian random number generator, with the variance of the generated Gaussian-distributed number at each grid cell equal to the NRCS value at the same grid cell. In this way, the speckle noise is incorporated into the 2D complex reflectivity field.

- Given the generated 2D wave height field, we can also calculate a 2D radial velocity field corresponding to large-scale sea waves with sizes greater than a grid cell using the transfer function relating the radial orbital velocities of large-scale sea waves to the wave heights [32,34]. Moreover, the spread of small-scale velocities within one single-grid cell is modeled as a random perturbation of the local long-wave orbital velocity [35]. In this way, the 2D random discrete radial sea wave velocity field can be realized.

In Step 2, the three obtained fields (i.e., 2D wave height, 2D complex reflectivity, and 2D radial sea wave velocity fields) are used as inputs and fed into a SAR raw data simulator called the extended Omega-K (IOK) algorithm [35], developed especially for ocean surface waves to obtain a SAR raw dataset for the moving sea surface. The following explanations are presented regarding the SAR raw data simulation:

- The 2D radial sea wave velocity field generated in Step 1 enters both the envelope and the phase of the 2D frequency (range and Doppler frequencies) complex spectrum of the SAR signal, from which the extended IOK algorithm developed in [35] can be implemented to simulate the SAR raw data for the moving sea surface. To account for the spatial variation of the ocean motion parameters, this simulator adopts the batch-processing operation, where a single implementation of the IOK algorithm will simultaneously simulate a collection of ocean surface backscattering elements with the same radial velocity, significantly raising the computational simulation efficiency. Further details about this simulator can be found in [35].

- A sinc-squared function is assumed for the Doppler envelope of the 2D frequency complex spectrum of the SAR signal.

- SAR raw data are first simulated with a relatively large PRF (e.g., twice the originally prescribed PRF). The simulated SAR raw data are then subsampled in the azimuth time domain by a factor of two to obtain the Doppler-aliased SAR raw data.

- A discrete grid of white thermal noise is added to the simulated SAR raw data such that a certain prescribed SNR is achieved. The prescribed SNR is determined as the ratio of the mean NRCS of the sea to the NESZ using Equation (14). The mean NRCS is computed according to a geophysical model function (GMF) (e.g., the CMOD5 GMF [36], the XMOD2 GMF [37], etc.), which directly relates the wind speed to the NRCS.

In Step 3, the SAR raw dataset simulated in Step 2 is processed using Equation (1) to obtain an estimate of the SAR Doppler centroid. In Step 4, independent Monte Carlo runs are performed, in which Steps 1–3 are repeated for $N_{MC}$ times using the same set of radar system and sea environmental parameters to obtain $N_{MC}$ Doppler centroid estimates, with $N_{MC}$ being an integer greater than at least 100. Finally, in Step 5, a statistical analysis is performed on these $N_{MC}$ Doppler centroid estimates to obtain their estimation STD and bias.

### 3.2. Example of Monte Carlo Simulations

The simulation results are provided to demonstrate how the Monte Carlo simulation method outlined in Figure 6 works using the specific set of SAR system and sea state parameters listed in Table 1. It should be noted that the mean NRCS ($-12$ dB) was determined by the XMOD2 GMF [37] for the incidence angle of 45°, the relative wind direction of 45°, and the wind speed of 13 m/s. Based on these parameters, a PM wave height spectrum was computed and shown as contour plots in Figure 7, where the color bar represents the spectral density. Implementing Step 1 in Figure 6 yielded the 2D wave height, 2D NRCS, and 2D radial sea wave velocity fields illustrated in Figures 7–9, respectively. The wave patterns of a fully developed wind sea are observed in the figures. The noisy characteristic observed in the 2D radial sea wave velocity field (Figure 10) originates from the random distribution of small-scale radial velocities within one single-grid cell. The RMS radial velocity of the wind sea measured from the resultant 2D radial sea wave velocity field (Figure 10) is 0.469 m/s, which is highly consistent with the value computed using Equation (23) (i.e., 0.4875 m/s). This result partially justifies the procedure of the sea wave velocity field generation shown in Figure 6.

**Table 1.** Radar system and sea state parameters.

| Parameter | Value |
|---|---|
| Radar platform velocity | 7600 m/s |
| NESZ | $-20$ dB |
| Antenna length in azimuth | 9.6 m |
| Chirp bandwidth | 40 MHz |
| Pulse repetition frequency | 1725 Hz |
| Doppler centroid estimation resolution | $1.0 \times 1.0$ km |
| Doppler bandwidth | 1403 Hz |
| Signal bandwidth | 40 MHz |
| Significant wave height | 2.8 m |
| Wind speed at a height of 10 m | 13 m/s |
| Incidence angle | 45° |
| Squint angle | 0° |
| Relative wind direction | 45° |
| Dominant wavelength | 155 m |
| Radar platform altitude | 700 km |
| Radar carrier frequency | 9.6 GHz |
| Range sampling rate | 80 MHz |
| Polarization | VV |
| Relative dielectric constant of ocean water | 48–35 J |
| Mean NRCS | $-12$ dB |
| Velocity component of current in azimuth | $-0.0$ m/s |
| Velocity component of current in ground range | $+0.65$ m/s |
| True Doppler centroid | $-29.3674$ Hz |
| Number of averaged pulses | 227 |
| Azimuth observation time | 0.1316 s |
| Number of averaged range samples | 380 |
| Number of independent Monte Carlo runs | 390 |

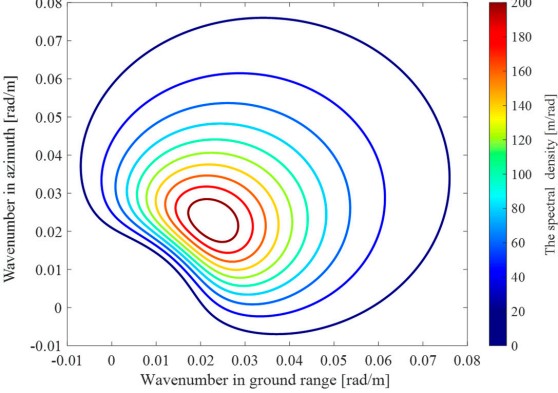

**Figure 7.** PM spectrum.

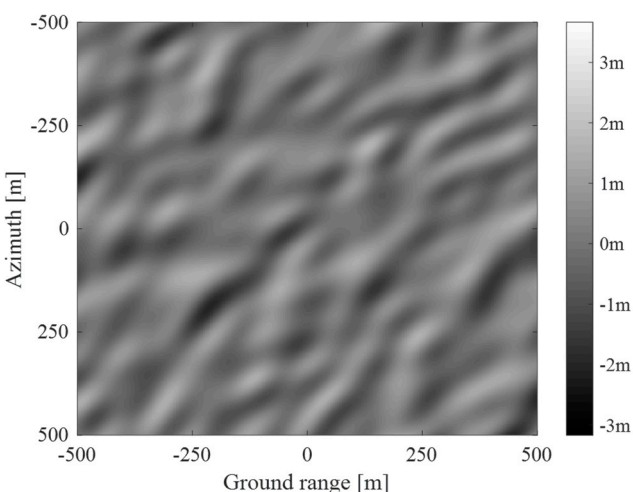

**Figure 8.** 2D wave height field.

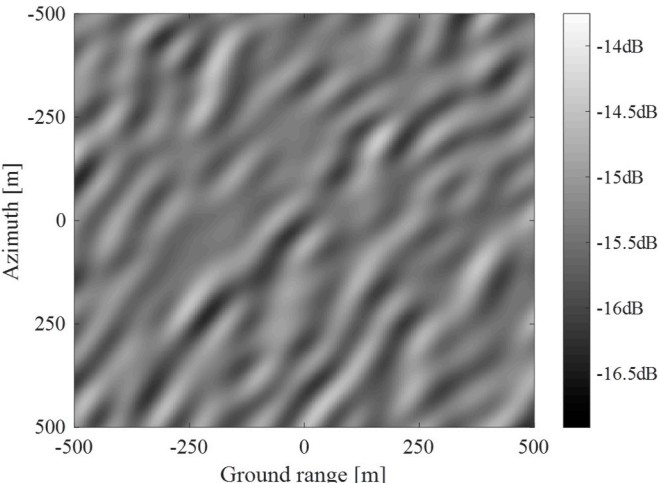

**Figure 9.** 2D NRCS field.

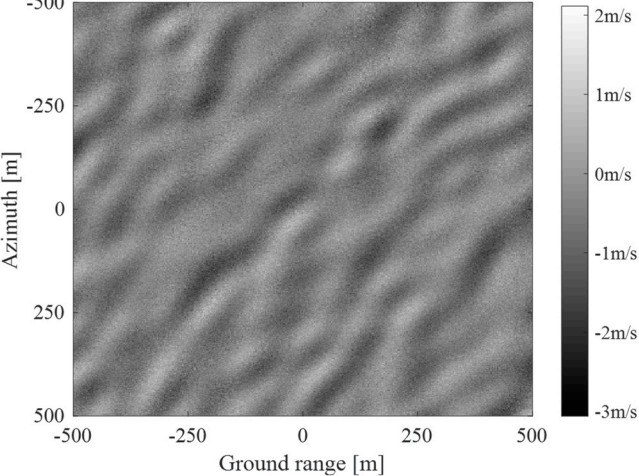

**Figure 10.** 2D radial sea wave velocity field.

Figure 11 depicts the amplitude of the 2D frequency spectrum of the complex SAR raw data obtained from Step 2 of Figure 6, clearly indicating its range frequency and Doppler envelopes. Figure 12 presents the SAR image obtained from processing the simulated SAR raw data. The wave-like patterns exhibited in Figure 12 are caused by the

tilt, hydrodynamic, range bunching, and velocity bunching modulations. Averaging the resultant 2D frequency power spectrum (Figure 11) along the range frequency dimension resulted in the one-dimensional Doppler power spectrum shown in Figure 13, which shows the speckle-and-thermal-noise-induced random fluctuations superimposed on a sinc-squared profile, with a shape consistent with that given in Equation (5).

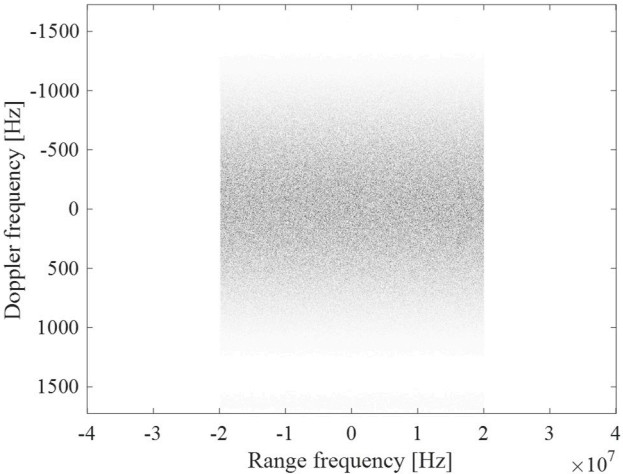

**Figure 11.** 2D frequency spectrum of the simulated SAR raw data.

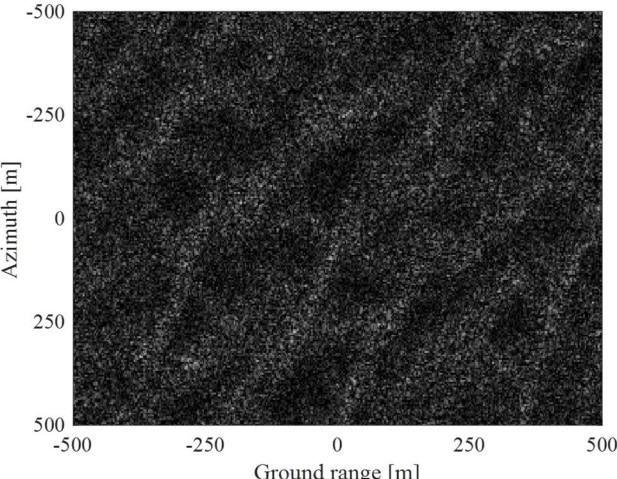

**Figure 12.** Processed SAR image.

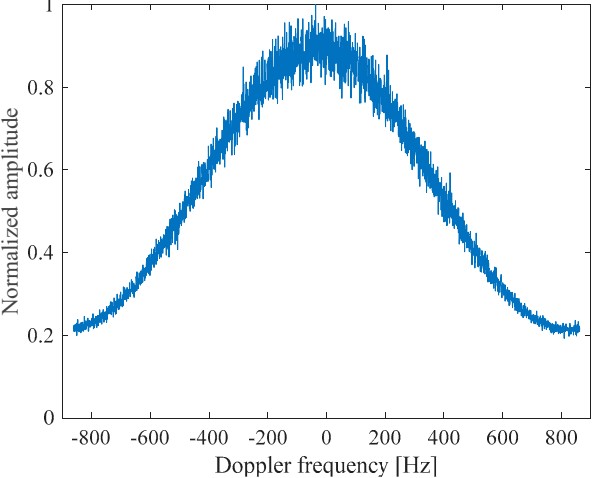

**Figure 13.** Fluctuation of Doppler power spectrum.

In this simulation example, 390 independent Monte Carlo simulation runs (Step 4 in Figure 6) were performed using the fixed set of the radar system and sea state parameters summarized in Table 1. A total of 390 SAR Doppler centroid estimates were obtained and shown as a scatter plot in Figure 14. This figure shows that these 390 Doppler centroid estimates keep fluctuating around the mean value of $-31.9343$ Hz (red line). Originating from the contributions of the wave-induced artifact Doppler velocities, the bias [17–19] between the measured mean Doppler centroid and the true value is calculated as 2.5669 Hz. The true value of the Doppler centroid is computed according to the function $f_{Dc} = -2v_r / \lambda$, where $v_r$ is the radial component of the current velocity vector for a zero-squint side-looking radar in our paper. The STD of these 390 Doppler centroid estimates is measured as 2.7668 Hz, whereas that computed from the newly derived formula [Equation (34)] in this study is 2.7891 Hz. The relative difference between the simulation-derived Doppler centroid estimation STD and the formula-predicted Doppler centroid estimation STD is 0.81%, which is low enough to indicate that the newly derived formula [Equation (34)] performs well in characterizing the STD of the SAR Doppler centroid estimates from a moving sea surface.

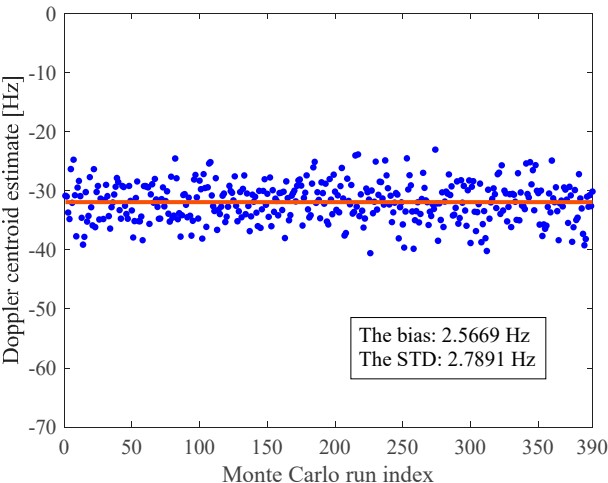

**Figure 14.** Scatter diagram of the Doppler centroid estimates obtained from 390 Monte Carlo runs. The bias and the STD of these 390 Doppler centroid estimates are annotated in the plot.

## 4. Comparison of the New Formula with Other Existing Formulas

In Section 3, we introduced the method of Monte Carlo simulations and provided a single case of SAR raw data simulation and Doppler centroid estimation for fixed radar system and sea state parameters. Here, we present a comprehensive comparison of the newly derived formula [Equation (34)] and other existing formulas to study their behavior in predicting the SAR Doppler centroid estimation STD against varying values of SAR system and sea state parameters, including the wind speed, the SNR, and the azimuth oversampling ratio.

The compared formulas include Bamler's formula (Equation (36), [23]) Liu's formula (Equation (18), [24]), our newly derived formula [Equation (34)], and a variant of Equation (34) with the effective number $N_r^{sea}$ [Equation (17)] of independent range samples of the sea wave velocity field replaced by the number of geometric range resolution cells. This variant of Equation (34) is included here to investigate whether the correlation length of the sea wave velocity field affects the SAR Doppler centroid estimation STD. The performances of the aforementioned formulas are compared using the method of Monte Carlo simulations for the radar and sea surface motions, and the specific simulation procedures are outlined in Figure 6 (more details can be seen in [35]).

*4.1. Doppler Centroid Estimation STD versus Wind Speed*

We first present the simulation results for the first scenario, in which the wind speed is varied over the 5 to 28 m/s interval, but the other radar system and sea state parameters shown in Table 1 remain unchanged. For each wind speed value, the corresponding NRCS is computed using the XMOD2 GMF [37]. For this scenario, 390 Monte Carlo runs are conducted at each wind speed, and a value of the Doppler centroid estimation STD is measured as described in the flowchart in Figure 6. Figure 15a plots the curve of the measured Doppler centroid estimation STD versus the wind speed. Superimposed on this plot are the curves of the Doppler centroid estimation STD predicted by the newly derived formula [Equation (34)], Bamler's formula (Equation (36), [23]), Liu's formula (Equation (18), [24]), and the variant of Equation (34), all against the wind speed.

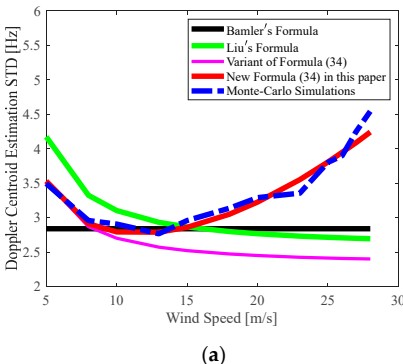
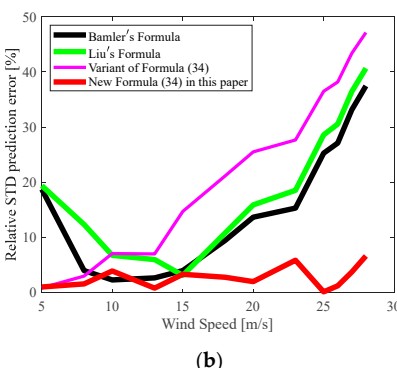

(a)　　　　　　　　　　　(b)

**Figure 15.** (**a**) Curves of the Doppler centroid estimation STD against wind speed; (**b**) Relative prediction error (in percentages), defined as the absolute value of the difference between the predicted and measured values of the Doppler centroid estimation STD divided by the latter.

Figure 15a shows that the Doppler centroid estimation STD measured from the Monte Carlo simulations first decreases with an increase in the wind speed until the wind speed reaches 13 m/s. Subsequently, the STD becomes larger with a further increase in the wind speed. This phenomenon is explained as follows: when the wind speed ranges from 5 to 13 m/s, the NRCS and, thus, the SNR of the sea surface increases with an increase in the wind speed. The Doppler centroid estimation STD then continuously becomes smaller. However, the role played by the sea wave motions becomes increasingly significant when the wind speed exceeds 13 m/s. The RMS radial velocity of the sea waves becomes larger with an increase in the wind speed [Equation (23)]. Simultaneously, the correlation length of the radial sea wave velocity field also becomes greater, indicating that the effective number of independent range samples decreases [see Equation (32)]. As a combined result of the aforementioned processes, the Doppler centroid estimation STD becomes larger.

From Figure 15, we make the following observations:

- Bamler's formula (Equation (36), [23]) fails to characterize the changes in the Doppler centroid estimation STD with the wind speed because it does not consider the SNR variation with the wind speed or the sea wave motions.
- Liu's formula (Equation (18), [24]) evidently underappreciates the effect of the sea wave motions on the Doppler centroid estimation STD when the wind speed exceeds 13 m/s because it fails to consider the correlation in the sea wave motion between two adjacent SAR range resolution cells. Instead, it takes the number of range resolution cells as the number of independent range samples of the sea wave velocity field, making the number of independent range samples of the sea wave velocity field used in this formula (Equation (18), [24]) larger than it is in practical situations.
- In contrast, the newly derived formula in this paper [Equation (34)] does not suffer from any of the aforementioned limitations. In Figure 15a, the curve of the Doppler centroid estimation STD predicted by the newly derived formula [Equation (34)] is more consistent with that obtained from the Monte Carlo simulations than the curves

predicted by Bamler's (Equation (36), [23]) and Liu's (Equation (18), [24]) formulas. The improvements in predicting the Doppler centroid estimation STD provided by the newly derived formula [Equation (34)] result from the consideration of both the SNR variation with the wind speed [see Equations (12) and (14)] and the variation of the correlation length of the sea wave velocity field with the wind speed [Equation (30)] derived in this work. These improvements can also be observed in Figure 14b. The relative prediction error, defined as the absolute value of the difference between the predicted and measured values of the Doppler centroid estimation STD divided by the latter, for the newly derived formula [Equation (34)] is smaller than that for Bamler's (Equation (36), [23]) and Liu's (Equation (18), [24]) formulas over most of the wind speed region.

- The significance of including the correlation length of the sea wave velocity field is further justified by observing the curve of the Doppler centroid estimation STD predicted by the variant of Equation (34), in which $N_r^{\text{sea}}$ [Equation (17)] is replaced by the number of geometric range resolution cells. In Figure 15, this curve exhibits poor agreement with that measured from the Monte Carlo simulations, especially for wind speeds above 10 m/s.

### 4.2. Doppler Centroid Estimation STD Versus SNR

For the second scenario, in which the radar system parameter NESZ, $\sigma_{\text{NE}}^0$, is varied from −50 to −8 dB, the same parameter setting shown in Table 1 is used, except for the NESZ. According to Equation (14), this NESZ variation region is equivalent to the SNR varied from −4 to 38 dB. For this scenario, the value of the Doppler centroid estimation STD is measured through 390 Monte Carlo runs at each SNR (Figure 6 in Section 3). Figure 16a shows the curves of the measured Doppler centroid estimation STD and those predicted by the newly derived formula [Equation (34)], Bamler's formula (Equation (36), [23]), Liu's formula (Equation (18), [24]), and the variant of Equation (34) against the SNR.

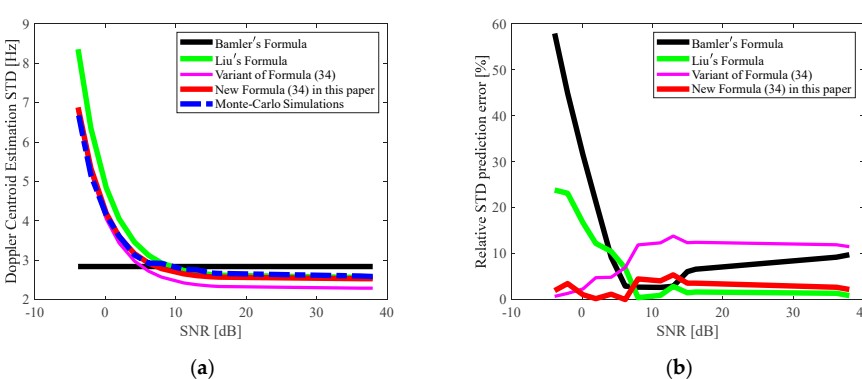

**Figure 16.** (**a**) Curves of the Doppler centroid estimation STD against the SNR; (**b**) Relative prediction error.

Figure 16a shows that the measured Doppler centroid estimation STD from the Monte Carlo simulations decreases with the SNR increase until the SNR reaches 10 dB. This is because the spectrum sharpness factor $m$ [Equation (12)] becomes larger as the SNR increases, meaning that the Doppler centroid can be determined with relative ease in the presence of speckles and thermal noises, and the Doppler centroid estimation STD then becomes continuously smaller. Figure 16a shows that, when the SNR exceeds 10 dB, the curve obtained from the Monte Carlo simulations flattens out. This is because once the SNR exceeds 10 dB, the spectrum sharpness factor $m$ nears its maximum and tends to converge, according to Equation (12), thereby exhibiting fewer effects on the Doppler centroid estimation STD. Consequently, the Doppler centroid estimation STD tends to be a fixed value.

Combining Figure 16a,b induces the following conclusions:

- Bamler's formula (Equation (36), [23]) cannot characterize the changes in the Doppler centroid estimation STD against the SNR because it sets its spectrum sharpness factor $m$ to a fixed value of 0.7 (Equation (28), [23]) without considering the SNR variation.
- Within the SNR range of $-4$ to 8 dB, the curve predicted by Liu's formula (Equation (18), [24]) is inconsistent with the curve obtained from the Monte Carlo simulations. This is because Liu's formula (Equation (18), [24]) heuristically quantifies the overall effect of the thermal noise and the Doppler aliasing on the Doppler centroid estimation STD as the product of their individual effects (Equation (A5), [24]). Consequently, this limits the ability of their formula to reflect the real situation.
- In Figure 16a, the curve of the Doppler centroid estimation STD predicted by the newly derived formula [Equation (34)] is more consistent with that obtained from the Monte Carlo simulations than those predicted by Bamler's (Equation (36), [23]) and Liu's (Equation (18), [24]) formulas. The improvement provided by the newly derived formula [Equation (34)] in predicting the Doppler centroid estimation STD is attributed to the inclusion of the SNR variation in Equation (34) and the effects of Doppler aliasing and thermal noise on the Doppler sharpness factor $m$ variations being jointly quantified in a mathematically exact, rather than heuristic, manner, as in Liu's formula (Equation (18), [24]). The improvements can also be observed from the relative prediction error versus the SNR, as shown in Figure 15b, for the newly derived formula [Equation (34)], which is more stable as a whole than those errors related to Bamler's (Equation (36), [23]) and Liu's (Equation (18), [24]) formulas.

### 4.3. Doppler Centroid Estimation STD Versus Azimuth Oversampling Ratio

For the third scenario, the along-track antenna length, $D_a$, is varied from 6 to 15 m, and the other parameters in Table 1 remain unchanged. Note that $D_a$ is a variable of the Doppler bandwidth function; thus, the variation region of $D_a$ is equivalent to the azimuth oversampling ratio, defined as $\gamma_{\mathrm{osr}}^{\mathrm{az}} = F_{\mathrm{prf}}/B_D$, varied from 0.7 to 1.9. For this scenario, a value of the Doppler centroid estimation STD is measured at each $\gamma_{\mathrm{osr}}^{\mathrm{az}}$ following the same procedures in Sections 4.1 and 4.2. Figure 17a shows the curves of the measured Doppler centroid estimation STD and those predicted by the newly derived formula [Equation (34)], Bamler's formula (Equation (36), [23]), Liu's formula (Equation (18), [24]), and the variant of Equation (34)) against $\gamma_{\mathrm{osr}}^{\mathrm{az}}$.

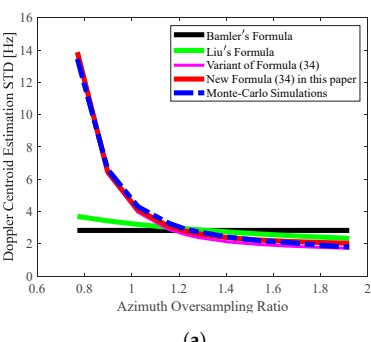
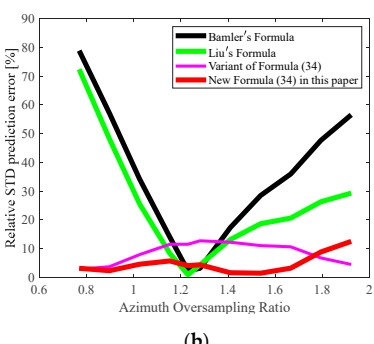

(a)                                                                 (b)

**Figure 17.** (**a**) Curves of the Doppler centroid estimation STD against the azimuth oversampling ratio; (**b**) Relative prediction error.

Figure 17a shows that the measured Doppler centroid estimation STD decreases sharply with an increase in $\gamma_{\mathrm{osr}}^{\mathrm{az}}$ before gradually leveling off when $\gamma_{\mathrm{osr}}^{\mathrm{az}}$ exceeds 1.2. This behavior is explained as follows: as $\gamma_{\mathrm{osr}}^{\mathrm{az}}$ increases from 0.7 to 1.2, the SAR Doppler bandwidth reduces, and thus the total amount of noise incorporated into the correlation results reduces; conversely, the spectrum sharpness factor $m$ [Equation (12)] increases. A combined consequence of the aforementioned processes induces the decrease in the SAR Doppler centroid estimation STD.

Based on a comparison of the curves in Figure 17, the following remarks can be made:

- Bamler's formula (Equation (36), [23]) visibly again fails to characterize the changes in the Doppler centroid estimation STD with the azimuth oversampling ratio because it uses a fixed azimuth oversampling ratio without considering the decoupling of the PRF and the Doppler bandwidth.

- Liu's formula (Equation (18), [24]) fails to accurately predict the variation of the Doppler centroid estimation STD with the azimuth oversampling ratio, though a mild change in the STD is found. This behavior can be explained as follows: Liu's formula (Equation (18), [24]) considers the effect of the Doppler bandwidth variation on the overall speckle noise level but does not account for its effect on the Doppler spectrum sharpness factor $m$ (this formula sets the Doppler aliasing-related spectrum sharpness factor to a fixed value of 0.7 regardless of its actual variation with the azimuth oversampling ratio).

- From Figure 17, the newly derived formula [Equation (34)] performs better than the other formulas in predicting the Doppler centroid estimation STD. These improvements are obtained because the newly derived formula fully decouples the PRF and Doppler bandwidth by considering the effect of the Doppler bandwidth variation on the overall speckle noise level [see Equation (7)] and expressing the spectrum sharpness factor $m$ as a function of the azimuth oversampling ratio [see Equation (12)]. This means that the PRF and Doppler bandwidth can independently take any value, as discussed in Section 2.1.

### 4.4. Overall Assessment of the Performance of the Newly Derived Formula

To assess the performance of the newly derived formula [Equation (34)] in predicting the Doppler centroid estimation STD further, we evaluated the average relative prediction errors over the variation regions of wind speed [Figure 14b], SNR [Figure 15b], and azimuth oversampling ratio [Figure 16b], for each of the aforementioned four formulas. These statistical quantities are listed in Table 2. The statistical measure of the correlation coefficient is adopted to quantify the degree of the trend correlation between the curves of the predicted and measured Doppler centroid estimation STDs. The correlation coefficient $\rho_{xy}$ is defined as follows:

$$\rho_{xy} = \frac{\sum_{i=1}^{N}(x_i - \overline{x})(y_i - \overline{y})}{\sqrt{\sum_{i=1}^{N}(x_i - \overline{x})^2} \cdot \sqrt{\sum_{i=1}^{N}(y_i - \overline{y})^2}}, \tag{35}$$

where $x_i$ and $y_i$ are the sample points of two variables, and $\overline{x}$ and $\overline{y}$ are their mean values. Table 2 presents the calculated correlation coefficients for the four formulas against wind speed, SNR, and azimuth oversampling ratio.

**Table 2.** Average relative prediction errors and correlation coefficients of different formulas.

| STD vs. Wind Speed | Average Relative Prediction Error (%) | Correlation Coefficient |
|---|---|---|
| Bamler's formula (Equation (36), [23]) | 16.10 | ≈0 |
| Liu's formula (Equation (18), [24]) | 19.10 | 0.2741 |
| Variant of Equation (34) | 22.69 | 0.2629 |
| Newly derived formula [Equation (34)] | 2.76 | 0.9764 |
| **STD vs. SNR** | **Average Relative Prediction Error (%)** | **Correlation Coefficient** |
| Bamler's formula (Equation (36), [23]) | 15.99 | ≈0 |
| Liu's formula (Equation (18), [24]) | 7.92 | 0.9987 |
| Variant of Equation (34) | 8.26 | 0.9988 |
| Newly derived formula [Equation (34)] | 2.61 | 0.9991 |
| **STD vs. Azimuth Oversampling Ratio** | **Average Relative Prediction Error (%)** | **Correlation Coefficient** |
| Bamler's formula (Equation (36), [23]) | 34.14 | ≈0 |
| Liu's formula (Equation (18), [24]) | 24.33 | 0.8653 |
| Variant of Equation (34) | 8.65 | 0.9993 |
| Newly derived formula [Equation (34)] | 4.69 | 0.9987 |

Table 2 shows that the average relative prediction error of the newly derived formula [Equation (34)] is always the smallest among the compared four formulas, whether against the wind speed, SNR, or azimuth oversampling ratio. This result means that the newly derived formula exhibits the highest average degree of agreement between the predicted and measured Doppler centroid estimation STDs. Furthermore, the correlation coefficients for the newly derived formula [Equation (34)] against different variables are above 0.9 and mostly larger than those for the other three formulas. In other words, the newly derived formula performs the best in capturing the variation trend of the measured Doppler centroid estimation STD among all four formulas. Note that the correlation coefficients of Bamler's formula (Equation (36), [23]) versus each variable all close to zero is due to the constant curve predicted by this formula; thus, it has a low vector similarity with the measured values obtained from the Monte Carlo simulations.

**5. Conclusions**

In this study, we derived a new formula [Equation (34)] and verified its effectiveness with the help of the Monte Carlo simulations. The results indicate that the Doppler centroid estimation STDs predicted by the new formula are in a better agreement with the measured values than those predicted by Bamler's (Equation (36), [23]) and Liu's (Equation (18), [24]) formulas, which is due to three contributions in this paper. First, we adopted a new strategy for determining the number of independent samples of the sea wave velocity field contributing to a Doppler centroid estimate by considering the correlation length in a range dictated by the ocean wavenumber spectrum to account for the effect of large-scale ocean wave motions. Second, the PRF and Doppler bandwidth were decoupled such that they (including the azimuthal oversampling ratio) can take any value in the newly derived formula [Equation (34)] to overcome the limitations in practical applications. Third, the effects of the thermal noise and Doppler aliasing were jointly quantified in a mathematically exact manner, improving the ability of the newly derived formula [Equation (34)] to reflect the actual situation.

Our study will provide a reference for developing a new radar system and system parameter design. In this study, we assume a fully developed wind sea, which means that our formula did not consider the effect of ocean swells. Furthermore, our formula has not been verified by the real SAR data. We consider leaving them as our future work.

**Author Contributions:** Conceptualization, S.Q.; methodology, S.Q. and B.L.; software, S.Q.; validation, S.Q. and B.L.; writing—original draft preparation, S.Q.; writing—review and editing, B.L.; supervision, B.L.; funding acquisition, B.L. and Y.H. All authors have read and agreed to the published version of the manuscript.

**Funding:** This research is supported in part by the National Natural Science Foundation of China, Grant numbers 42027805, 41620104003, and 41606201.

**Data Availability Statement:** The data presented in this study were generated by the simulation algorithm in [35].

**Conflicts of Interest:** The authors declare no conflict of interest.

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
