# Peer review of "Improved Analytical Formula for the SAR Doppler Centroid Estimation Standard Deviation for a Dynamic Sea Surface"

_remotesensing, doi:10.3390/rs15030867_

Round 1

Reviewer 1 Report

General comments:

This manuscript presents an improved formula for SAR Doppler centroid estimation standard deviation, and verifies its effectiveness with the help of the Monte-Carlo simulation. However, the novelty of the work is limited and the contribution of the work is not well organized. The English should be significantly improved. More importantly, the performance is only validated with simulation not real data. In conclusion, I suggest the paper be accepted after major revisions.

Specific Comments:

1.     Page 3 Line 127-128, “The SAR Doppler centroid is a measure of the effective radar beam squint angle”, why? To my knowledge, the Doppler centroid corresponds to a velocity not an angle.

2.     As shown in Equation (5) and Equation (8), why the expected Doppler power spectrum of the radar echoes can be expressed as a cosine and a sinc-squared, respectively. Please be more specific.

3.     Equation (10), why Pn is not a random quantity, but a fixed value?

4.     Does Section 4 provide the SAR raw data simulation for a moving radar and the random motions of sea surface waves? As shown in Page 4 Line 156, the component of the Doppler centroid estimation variance includes radar motion, and what is the effect of radar motion?

5.     As shown in Table 2, the newly derived formula in this paper performs well in capturing the variation trend of the measured Doppler centroid estimation STD of the Monte-Carlo simulation, how about the real radar data? In other word, does good performance mean that using this formula on real radar data to estimate the Doppler centroid can obtain a higher degree of confidence?

Reviewer 2 Report

The article is well and clearly written and contains useful information for improved SAR data processing. The article is a continuation of the authors' research and contains links to their previous works, which is also useful. There are also links to important works by other authors on this topic.

The RMS radial velocity should depend on the direction of sounding relative to the general propagation of the wave, as shown in (20). However, with further analysis, this dependence disappears. It turns out that the Doppler bandwidth does not depend on the direction of probing, which should not be the case. It may be erroneous to designate the relative angle between the sea wave propagation and radar flight directions and the propagation direction angle of the wave number vector k with the same symbol.

CMOD5 is used to calculate mean NRCS, and Table 1 shows the parameters of X-band - 9.6 GHz. Is there a mistake?

How was the True Doppler centroid value obtained?

Explain the colorbar designations in Figure 6.

You can skip the word "Equation" when referring to formulas given in the text.

Reviewer 3 Report

The advantages of An Improved Analytical Formula for the SAR Doppler Centroid Esti- 125 mation STD for a Dynamic Sea Surface are confirmed by numerical experiment using the Monte Carlo method. However, the description of the sea surface is rather simplified, in particular, it is assumed that the different wavelengths are independent of each other. Comparison of calculations using previously used formulas and the new formula demonstrates a very significant difference. It seems that the authors should present data from satellite experiments, synchronous with the measurement of the sea surface characteristics, which would demonstrate the shortcomings of the previous formulas and the advantages of the new one.

Round 2

Reviewer 3 Report

My comment was that the need for the study and the correctness of the obtained results were only confirmed by numerical simulation. No experimental results were given. The authors promised to do that in a future paper. Let it be so. 

Author Response

Dear Reviewer,

We would like to express our sincere gratitude to you for your interest in our paper and all the efforts and time spent on our manuscript as well as for so valuable remarks and suggestions on our manuscript! We believe your comments and suggestions are very helpful in improving the quality of our paper. Thank you again.

Sincerely yours,

Siqi Qiao and Baochang Liu